# Behavior-Aware Pedestrian Trajectory Prediction in Ego-Centric Camera Views with Spatio-Temporal Ego-Motion Estimation †

**Phillip Czech** [1,2,*] , **Markus Braun** [1] , **Ulrich Kreßel** [1] and **Bin Yang** [2]

[1] Perception & Maps Department, Mercedes-Benz AG, 71063 Sindelfingen, Germany; markus.ma.braun@mercedes-benz.com (M.B.); ulrich.kressel@mercedes-benz.com (U.K.)

[2] Institute of Signal Processing and System Theory, University of Stuttgart, 70550 Stuttgart, Germany; bin.yang@iss.uni-stuttgart.de

[*] Correspondence: phillip.czech@mercedes-benz.com

[†] This paper is an extended version of our paper published in the Proceedings of the 21st IEEE International Conference on Machine Learning and Applications (ICMLA), Nassau, Bahamas, 12–14 December 2022.

**Abstract:** With the ongoing development of automated driving systems, the crucial task of predicting pedestrian behavior is attracting growing attention. The prediction of future pedestrian trajectories from the ego-vehicle camera perspective is particularly challenging due to the dynamically changing scene. Therefore, we present Behavior-Aware Pedestrian Trajectory Prediction (BA-PTP), a novel approach to pedestrian trajectory prediction for ego-centric camera views. It incorporates behavioral features extracted from real-world traffic scene observations such as the body and head orientation of pedestrians, as well as their pose, in addition to positional information from body and head bounding boxes. For each input modality, we employed independent encoding streams that are combined through a modality attention mechanism. To account for the ego-motion of the camera in an ego-centric view, we introduced Spatio-Temporal Ego-Motion Module (STEMM), a novel approach to ego-motion prediction. Compared to the related works, it utilizes spatial goal points of the ego-vehicle that are sampled from its intended route. We experimentally validated the effectiveness of our approach using two datasets for pedestrian behavior prediction in urban traffic scenes. Based on ablation studies, we show the advantages of incorporating different behavioral features for pedestrian trajectory prediction in the image plane. Moreover, we demonstrate the benefit of integrating STEMM into our pedestrian trajectory prediction method, BA-PTP. BA-PTP achieves state-of-the-art performance on the PIE dataset, outperforming prior work by 7% in MSE-1.5 s and $C_{MSE}$ as well as 9% in $CF_{MSE}$.

**Keywords:** pedestrian trajectory prediction; autonomous driving; behavioral features; ego-motion compensation

## 1. Introduction

One of the essential challenges for automated driving in urban traffic conditions is the task of pedestrian behavior prediction. Therefore, understanding the underlying intentions of pedestrians is crucial for automated vehicles to better understand their surroundings in order to make better and safer driving decisions and mitigate potential risks and hazardous scenarios [1]. Due to the inherently variable nature of pedestrians' behavior, solving the task of predicting this behavior involves numerous challenges. Nonetheless, human drivers are able to anticipate the future actions of pedestrians by relying on visual observations and interpreting their behavioral cues. Inferring the potential future movement direction of pedestrians based on their body or head orientation and responding appropriately is a suitable example.

In this work, we focused on the problem of pedestrian trajectory prediction from an ego-centric perspective captured by a camera mounted on board a vehicle, e.g., behind

the windshield. Hereby, the objective is to predict the future bounding boxes in the image plane. To accomplish this, it is necessary to extract rich information from camera-based observations within a remarkably dynamic environment. The majority of current state-of-the-art approaches for predicting future bounding boxes heavily depend on the past positions of pedestrians. These are utilized to encode their motion history [2,3] or condition the predictions based on estimated inherent goals [4,5]. Taking into account the past trajectory of pedestrians is one of the most important features for this prediction task and is particularly advantageous in scenarios where they exhibit linear motion patterns. However, pedestrians consistently adapt their intended paths due to the influence of the constantly evolving environment surrounding them. For example, a pedestrian may all of a sudden decide to cross the street after standing on the sidewalk for some time. In such scenarios, additional information beyond just the past trajectory is required to accurately predict the future path of pedestrians, especially in ego-centric camera views where the scene dynamically changes due to the ego-motion of the vehicle. By observing pedestrians from the perspective of an ego-vehicle camera, it becomes possible to capture behavioral features that provide valuable cues about their intended movement. These cues can be extracted from visual observations and explicitly modeled through features like the body and head orientation as well as their pose. Despite that, behavioral features inferred from visual observations are only used in a few works [6–9] to predict pedestrians' future bounding boxes.

In this work, we proposed a pedestrian trajectory prediction method for ego-centric camera systems that uses behavioral features such as body orientation and head orientation, as well as pedestrian's pose. The different input modalities are processed by independent encoding streams and the resulting encodings are fused to better model the motion history of pedestrians. By visually observing pedestrians, we make use of the cues they provide about their intended movements and incorporate them explicitly into our method. We refer to the method as **Behavior-Aware Pedestrian Trajectory Prediction (BA-PTP)**. An overview of our proposed method is shown in Figure 1.

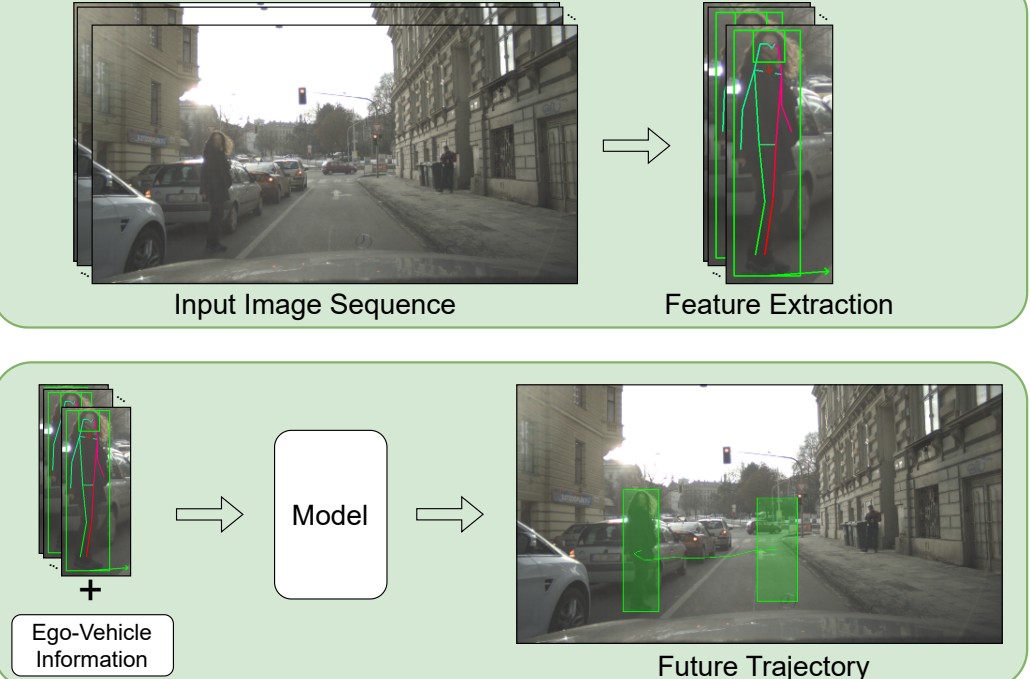

**Figure 1.** Our proposed pedestrian trajectory prediction method (BA-PTP) utilizes behavioral features such as body and head orientation as well as pose in addition to body and head bounding boxes. These features are combined with ego-vehicle information to predict the future bounding boxes of a pedestrian in the image.

When predicting pedestrian trajectories in the image plane, there is a need to compensate for the ego-motion of the camera in this dynamically changing scene. Current approaches tackle this by incorporating estimated odometry information of the ego-vehicle into the trajectory prediction [2,3,10]. However, these approaches solely rely on past information without utilizing the future intended route of the ego-vehicle.

In contrast to that, we integrated a module into BA-PTP, which leverages spatial goal points from the intended route of the ego-vehicle for predicting its future ego-motion, called **Spatio-Temporal Ego-Motion Module (STEMM)**. This route can be provided, e.g., by a navigation system. STEMM follows a similar scheme as the bounding box prediction, whereas it encodes the ego-vehicle's past odometry and utilizes the goal points to predict future ego-motion. The predictions are used by BA-PTP to predict future pedestrian trajectories.

**Contributions:** (1) We propose BA-PTP, a novel method for pedestrian trajectory prediction in the image plane, which incorporates behavioral features of pedestrians using independent encoding streams. (2) We introduce STEMM, a novel approach for predicting the future ego-motion of the ego-vehicle using spatial goal points, which is integrated into BA-PTP. (3) We performed a detailed analysis of BA-PTP on two pedestrian behavior datasets, demonstrating the advantages of incorporating behavioral features as well as STEMM into a pedestrian trajectory prediction model. (4) Based on ablation studies, we evaluated the effectiveness of our proposed encoding strategy and explored the impact of various behavioral features on the performance of prediction.

Preliminary results of this research are published in a conference paper at ICMLA 2022 [11]. In this work, we extended this in the following ways:

- We improved the performance of the BA-PTP method from [11] by focusing on a more stable encoding of pedestrian trajectories in the image plane. We did this by first incorporating information about the head position of pedestrians into our method. Compared to the body bounding boxes of a pedestrian, this modality is not prone to changes in shape. Second, by adapting the bounding box representation compared to our preliminary work.
- To compensate for the ego-motion in pedestrian trajectory prediction from the ego-vehicle camera perspective, we integrated STEMM into BA-PTP. Compared to our previous work, this replaces the use of precise temporal future ego-vehicle odometry information. In a real-world application with unknown future odometry, spatial goal points may be extracted from the intended route provided by a navigation system.
- BA-PTP achieves a state-of-the-art performance on the PIE dataset, outperforming prior work in MSE-1.5s, $C_{MSE}$, and $CF_{MSE}$.

This work is organized as follows: It begins with reviewing related works in Section 2. In Section 3, we present BA-PTP and STEMM as an approach to predicting future trajectories of pedestrians in the image plane. Then, we show experimental results and ablation studies in Section 4, before discussing our findings and elaborating on future research directions in Section 5. Finally, we conclude this work in Section 6.

## 2. Related Work

The field of pedestrian behavior prediction has witnessed increasing research attention in recent times. Predicting pedestrians' future behavior is approached through two distinct modeling methods, which differ in terms of their output. The first approach involves the prediction or classification of the *future action* of a pedestrian, i.e., whether he will cross the street in the near future, often referred to as intention prediction [7,12–25]. In the second approach, the focus lies on predicting the *future trajectory* of a pedestrian for a defined prediction horizon [2–7,10,14,26–35]. The former approach primarily provides insights solely into a potential crossing action occurring at some point in the future, while the latter predicts positional information that can be utilized to derive the crossing action as well.

Moreover, methods working on the problem of trajectory prediction can be distinguished by the perspective in which they operate. Pedestrian trajectory prediction from

the ego-vehicle camera perspective observes the environment from a moving first-person view. This results in a dynamically changing scene, because the relative position and size of observed objects depend on the ego-motion of the camera, as the ego-vehicle is also moving. To address this, some methods try to compensate for the ego-motion of the camera by incorporating ego-vehicle information [2,3,6,8–10,14,22,34,35].

Most of these approaches try to tackle this problem by predicting the future odometry of the ego-vehicle [2,3,6]. They first encode the history of the ego-vehicle's odometry using a Recurrent Neural Network (RNN) encoder. Sometimes, additional information, such as action and interaction priors [10] or visual features [3], is also incorporated into the encoding. Subsequently, the encoded data are fed into an RNN decoder for predicting the future ego-vehicle odometry. Similar to the aforementioned approaches, ego-motion compensation is also performed by simply encoding the observed ego-vehicle's odometry [8–10,14,22] or combining this with optical flow representations [35]. Ref. [34] proposed a method that explicitly disentangles the motion of pedestrians and the ego-motion of the vehicle. This is accomplished by using an ego-motion prediction network to observe and predict pedestrian behavior from an ego-motion compensated viewpoint.

When predicting the trajectory from a bird's eye perspective, the environment is observed from a top-down view, where the positions of objects are represented by global coordinates and are not dependent on the ego-motion of the camera. This perspective allows for better modeling of interactions [29–32] as relative distances between objects can be inferred from the global coordinates. Recently, this has also been investigated in the ego-centric camera view domain [20,36,37]. These methods try to model the relationships between the target pedestrian and nearby objects in the spatial and temporal domain by constructing scene graphs using object locations in the image plane.

Nowadays, recurrent architectures [2,3,15], as well as CVAEs [4,5,26], are prevalent in state-of-the-art pedestrian behavior prediction and are used in the majority of recent works. Other approaches model spatio-temporal features by first extracting features using CNNs [10,12,15] or graph structures [6,13,20] and processing them later on with RNNs. Recently, partially attention-based [2,15,23,24] transformers [7,8,25,35,38], as well as goal-driven [4,5,9,33] approaches, have also gained growing interest.

Related work may also be differentiated by the information used as input to predict the future behavior of pedestrians. The pedestrian's motion history is the most fundamental information as almost all methods rely on the past motion of the pedestrian independent of the domain [2–10,12–16,22–34,36,37], This information includes bounding boxes for ego-centric camera view methods or global coordinates for methods in bird's eye view. Additionally, features like the distance to the ego-vehicle [26,27], semantic and contextual information [7,13,14,20,23,24,26,27,31,32,36], as well as visual or appearance features [2,3,6,10,12,15–17,19,20,23–25,34,36,37], are used as inputs to encode the past motion and behavior of pedestrians.

Behavioral cues such as body or head orientation [7,8,13,26,28], awareness [8,17,21,27], pose of the pedestrian [6,8,9,15,18,19,22–24], and pedestrians' intention as the wish to cross the street [2,6,16] are also used to better understand and model pedestrians' behavior.

However, in the domain of pedestrian trajectory prediction from the ego-vehicle camera perspective, such behavioral features are rarely used. [6] used pedestrians' poses to improve the prediction of their intention module, resulting in only a marginal benefit for the trajectory prediction on PIE [2]. The body orientation of pedestrians was used in [7] to simultaneously predict intention and trajectory, but only benefits intention prediction, whereas this reduces trajectory prediction performance.

After the initial submission of our preliminary work [11], two further related works have been published [8,9]. Ref. [8] proposed an approach, where different modalities—namely pedestrian bounding boxes, their poses, and the ego-motion of the vehicle—are encoded using crossmodal transformers and concatenated with an embedding of a pedestrian attribute vector, which contains the pedestrian's body orientation, bounding box and its awareness of the ego-vehicle as well as the vehicle ego-motion of the last observed

timestep. This attribute vector should not only encode the basic state of the pedestrian but also implicitly the relationship between the pedestrian and the ego-vehicle [8]. Pedestrian poses are also used in the goal-driven approach of [9] in addition to bounding boxes and the ego-vehicle's motion. Moreover, the authors made use of ground truth future odometry information to compensate for the ego-motion of the vehicle.

In contrast to the methods published before our preliminary work, we show how to incorporate behavioral features, such as the body and head orientation of pedestrians, as well as their pose, to benefit the prediction of future trajectories in the image plane. Further, we included information about the head position of pedestrians in our method to better encode the trajectories of pedestrians. We used independent encoding streams for each input modality and fused the learned embeddings with an attention mechanism to better encode the motion history of pedestrians and predict more accurate future trajectories. Additionally, we integrated a module that predicts the camera's ego-motion into our trajectory prediction approach to account for the dynamically changing scene in ego-centric views. Compared to previous works, we utilized spatial goal points, that were sampled from the intended route of the ego-vehicle. Table 1 demonstrates a characteristic comparison of our approach against prior works for pedestrian trajectory prediction in ego-centric camera views, including our preliminary work [11].

**Table 1.** Characteristic comparison of our approach against prior works for pedestrian trajectory prediction in ego-centric camera views. We compared in terms of which pedestrian features are used (BB = Body Bounding Box, HB = Head Bounding Box, P = Pose, BO = Body Orientation, HO = Head Orientation), whether predicted ego-motion is incorporated for bounding box prediction, and if the predictions are conditioned on estimated goals.

| Method | Pedestrian Features | | | | | Ego-Motion Prediction | Goal-Driven |
| | BB | HB | P | BO | HO | | |
| --- | --- | --- | --- | --- | --- | --- | --- |
| Bhattacharyya et al. [3] | ✓ | ✗ | ✗ | ✗ | ✗ | ✓ | ✗ |
| Rasouli et al. [2] | ✓ | ✗ | ✗ | ✗ | ✗ | ✓ | ✗ |
| Yao et al. [5] | ✓ | ✗ | ✗ | ✗ | ✗ | ✗ | ✓ |
| Wang et al. [4] | ✓ | ✗ | ✗ | ✗ | ✗ | ✗ | ✓ |
| Fu et al. [9] | ✓ | ✗ | ✓ | ✗ | ✗ | ✗ | ✓ |
| Su et al. [8] | ✓ | ✗ | ✓ | ✓ | ✗ | ✗ | ✗ |
| Czech et al. [11] | ✓ | ✗ | ✓ | ✓ | ✓ | ✗ | ✗ |
| **Ours** | ✓ | ✓ | ✓ | ✓ | ✓ | ✓ | ✗ |

## 3. Method

We define the task of forecasting pedestrian behavior as the prediction of future trajectories in terms of image positions. This prediction is based on real-world traffic scene observations obtained from the ego-vehicle camera perspective. Hereby, pedestrians' trajectories are represented as a sequence of body bounding boxes in the image plane. However, these sequences of body bounding boxes are prone to fluctuations in their box widths due to the natural gait cycle of humans, which can be referred to as *box wobbling*. Figure 2 shows exemplary trajectories of (crossing) pedestrians, which show how the bounding box width changes during the observation. To overcome this downside and get a more stable representation of pedestrian trajectories, we proposed to also include information about the head position of pedestrians. Head bounding boxes do not have the property of changing shapes and can thus help to better encode the motion history of pedestrians and reduce the box wobbling effects. This can also be seen in Figure 2.

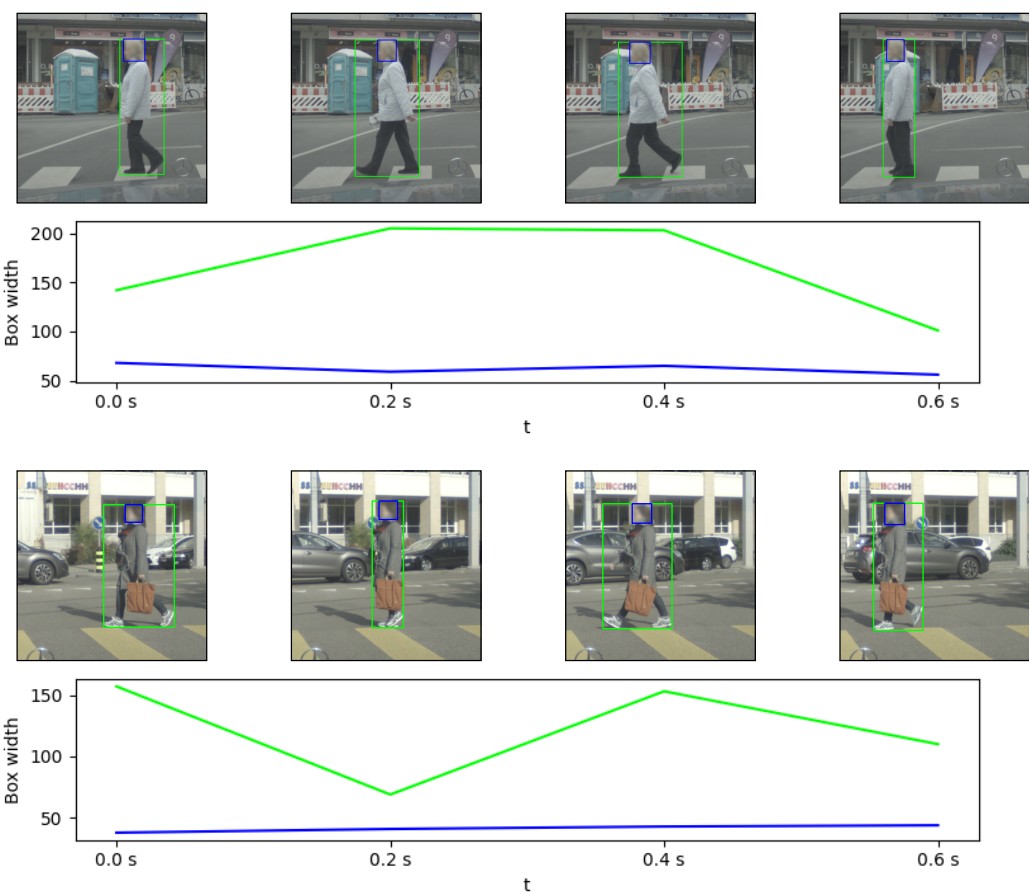

**Figure 2.** Visualization of box wobbling. We display cropped images from consecutive timesteps with body (green) and head (blue) bounding boxes of crossing pedestrians at a frame rate of 5 fps. Moreover, we plotted the respective box width over time. This figure demonstrates the fluctuations of body bounding box widths due to the natural gait cycle of humans. In contrast to that, the head bounding boxes are not prone to changes in their shapes.

Bounding boxes are usually represented by either the top-left and bottom-right corners in pixel coordinates in the image $bb = \{x_{tl}, y_{tl}, x_{br}, y_{br}\}$, or the center coordinate as well as width and height in pixels: $bb = \{c_x, c_y, w, h\}$. Our previous publication [11] used the first representation. Here, we switched to the latter representation to further reduce box wobbling effects.

We aimed to predict the future trajectory $\text{Traj}_{fut} = \{bb_i^{t+1}, \ldots, bb_i^{t+m}\}$ of a pedestrian $i$ at a certain timestep $t$ for a prediction horizon $m$, given the pedestrian's observed past trajectory represented by body bounding boxes $BB_{obs} = \{bb_i^{t-n+1}, \ldots, bb_i^t\}$ and head bounding boxes $HB_{obs} = \{hb_i^{t-n+1}, \ldots, hb_i^t\}$ for an observation horizon of length $n$, as well as observed behavioral features of the pedestrian:

- Body orientation $BO_{obs} = \{bo_i^{t-n+1}, \ldots, bo_i^t\}$, where $bo \in \mathbb{R}$;
- Head orientation $HO_{obs} = \{ho_i^{t-n+1}, \ldots, ho_i^t\}$, where $ho \in \mathbb{R}$;
- Pose $P_{obs} = \{p_i^{t-n+1}, \ldots, p_i^t\}$, where $p \in \mathbb{R}^{34}$ defines the pixel coordinates of a 17-joint skeleton.

Additionally, we made use of the predicted future ego-motion of the ego-vehicle $EM_{fut} = \{em^{t+1}, \ldots, em^{t+m}\}$, where $em^{t+j} \in \mathbb{R}^{d_{em}}$ is a vector with dimensionality $d_{em}$. The ego-motion vectors are predicted by a separate module, which is introduced in Section 3.1.2. In different experiments, the predicted ego-motion of this module is either speed and yaw rate (i.e., $d_{em} = 2$) or an arbitrary feature embedding.

*3.1. Architecture*

Pedestrians implicitly offer numerous cues regarding their intended movement as they walk along a street. These cues can be inferred through visual observations and expressed as behavioral features, such as the body and head orientation of pedestrians as well as their pose. To account for this, we proposed **Behavior-Aware Pedestrian Trajectory Prediction (BA-PTP)**, a method that uses this information as explicit features in addition to their motion history. Furthermore, our novel approach for ego-motion prediction, called **Spatio-Temporal Ego-Motion Module (STEMM)**, provides predicted future ego-motion information to be utilized by BA-PTP.

Initially, we present our BA-PTP architecture, followed by the introduction of STEMM.

3.1.1. Pedestrian Trajectory Prediction

As depicted in Figure 3, BA-PTP consists of an RNN encoder-decoder architecture, where we process each input modality through an independent encoder, namely a body bounding box stream, a head bounding box stream, a pose stream, a head orientation stream, and a body orientation stream.

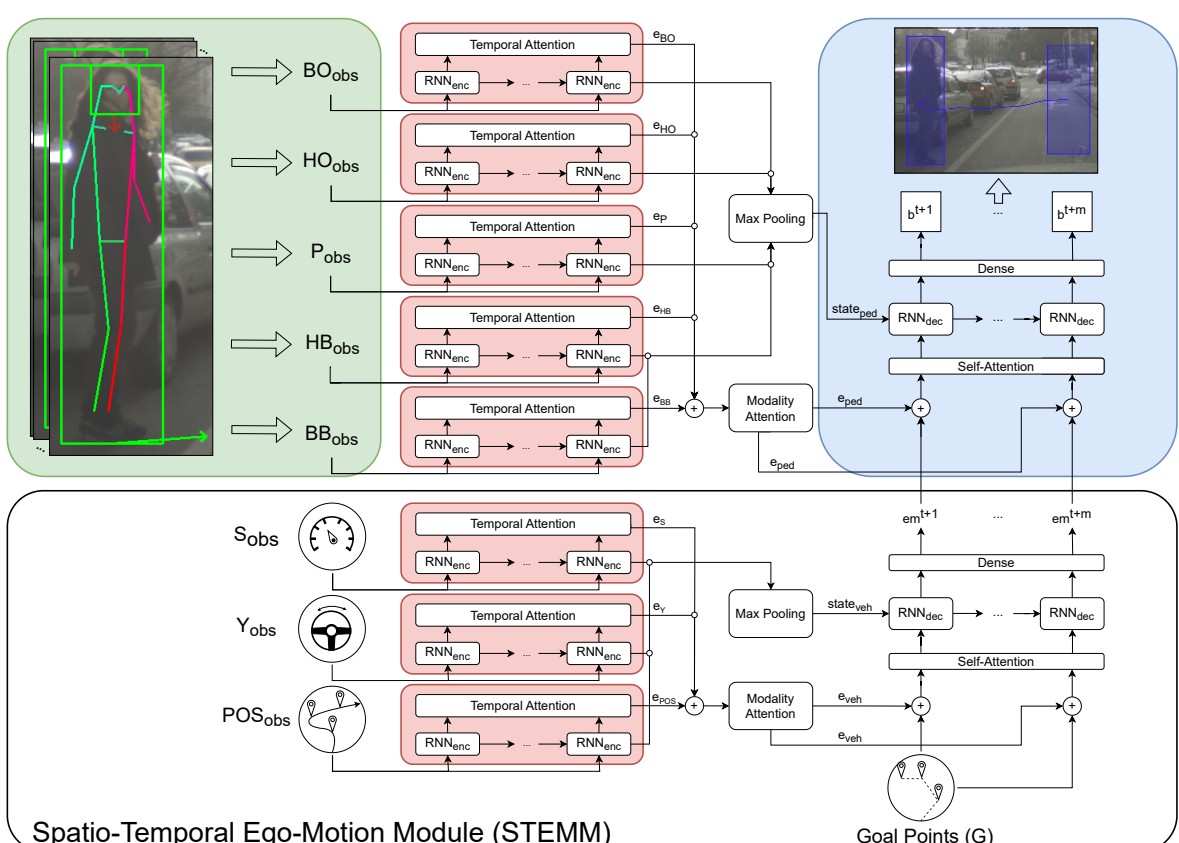

**Figure 3.** Diagram of our proposed method BA-PTP. The inputs to the method are body and head bounding boxes and behavioral features, such as pedestrian pose and head and body orientation for a defined observation horizon. Each input modality is processed by an independent encoding stream and the outputs of each stream are fused using a modality attention module, resulting in a final embedding vector. For each prediction timestep, we concatenated the final embedding vector with future ego-motion information and passed this through a self-attention unit. This is then used as the decoder inputs to predict future bounding boxes in the image. For predicting the future ego-motion of the ego-vehicle, we integrated STEMM into BA-PTP. STEMM follows a similar scheme as the bounding box prediction, but the inputs are speed, yaw rate, and ego-vehicle positions from the observation horizon. Moreover, STEMM utilizes spatial goal points sampled from the intended route to predict the future ego-motion of the ego-vehicle.

The encoding scheme of BA-PTP is inspired by [15]. We applied the same structure for each encoding stream $e$ as follows: The RNN encoder processes the input sequence and produces hidden states for each observed timestep. We then applied the temporal attention mechanism from [15] on the hidden states. The role of the attention mechanism is to give higher importance to particular timesteps in the observation horizon compared to other ones, focusing on the most relevant information. The resulting embedding vectors $e_e$ of each encoding stream are concatenated and the modality attention module from [15] produces the final embedding vector $e_{ped}$ by assigning importance to the different modality inputs.

The final embedding created by our encoding module contains a latent representation of the motion history of the pedestrian, which can be used to predict the future trajectory. To get a joint representation as input to the trajectory predictor, we concatenated the embedding vector $e_{ped}$ with ego-motion information $em^{t+j}$ for every future timestep $t + j$. We passed this through a self-attention unit from [2] to target the features of the encoding that are most relevant for the present prediction task. We then used this as the decoder inputs, resulting in hidden states for every future timestep in the prediction horizon. To generate the final predictions, we passed the decoder outputs through a dense layer that regresses the hidden states for every future timestep into bounding box predictions $\text{Traj}_{fut} = \{bb_i^{t+1}, \ldots, bb_i^{t+m}\}$.

We calculated the decoder's initial state vector $state_{ped}$ by taking the elementwise maximum (max-pooling [39]) of the final hidden states $h_e^t$ of all encoding streams, with $e$ being the index of an encoding stream:

$$state_{ped} = \max_{1 \leq j \leq d} h_{ej}^t, \tag{1}$$

where $d$ is the dimensionality of the hidden states. We found this to result in the best performance.

3.1.2. Ego-Motion Prediction

In this work, we proposed Spatio-Temporal Ego-Motion Module (STEMM), a novel approach for predicting the future ego-motion of the ego-vehicle. The main contribution of STEMM compared to other approaches for odometry prediction, as described in Section 2, is the incorporation of spatial goal points of the ego-vehicle. These goal points represent target locations along the route to be followed, e.g., provided by a navigation system. This approach is inspired by recent methods for planning in the context of autonomous driving that also use goal points to describe the driver intention or the route to be followed [40–42]. For STEMM, the goal points should provide priors about where the ego-vehicle might move to better predict the future odometry that follows the intended path. Inspired by [41], a goal point $g = \{g_x, g_y, g_{yaw}\}$ is represented by the x- and y-coordinates, $g_x$ and $g_y$, as well as the yaw angle $g_{yaw}$ of the ego-vehicle in a bird's eye view and ego-centric coordinate system. As the used datasets in this work do not provide the intended route from the navigation system itself, we approximate it using odometry information. Details are described in Section 4.2.

Goal points are sampled using a simple yet effective approach. Due to the fact that we do not have precise temporal future odometry information at present, last observed, timestep $t$, we sample spatial future goal points from the intended route to be followed by the ego-vehicle. We utilized the present velocity of the ego-vehicle to estimate its distance traveled during the prediction horizon $m$, assuming constant velocity to calculate the spatial distance $dist_g^{t+j}$ for a future timestep $t + j$. We obtain $g^{t+j}$ by extracting the point with spatial distance $dist_g^{t+j}$ from the intended route of the ego-vehicle. In total, we sample $m$ goal points $G = \{g^{t+1}, \ldots, g^{t+m}\}$, one for each future timestep. STEMM is named for its use of *spatial* information to predict *temporal* features.

In addition to the set of goal points $G$, the input to STEMM is the ego-vehicle odometry information for the $n$ observed timesteps:

- Speed $S_{obs} = \{s^{t-n+1}, \ldots, s^t\}$;
- Yaw rate $Y_{obs} = \{y^{t-n+1}, \ldots, y^t\}$;
- Past ego-vehicle positions $POS_{obs} = \{pos^{t-n+1}, \ldots, pos^t\}$, with $pos = \{pos_x, pos_y, pos_{yaw}\}$ similar to goal point $g$.

STEMM predicts the future ego-motion of the ego-vehicle $EM_{fut} = \{em^{t+1}, \ldots, em^{t+m}\}$, where the ego-motion vector $em^{t+j} \in \mathbb{R}^{d_{em}}$ for timestep $t + j$ has dimensionality $d_{em}$.

STEMM follows a similar scheme as the bounding box prediction, whereas the ego-vehicle's odometry history is encoded using a speed stream, a yaw rate stream, and a past ego-vehicle position stream. The architecture of STEMM is illustrated in the lower part of Figure 3. The final embedding $e_{veh}$ is concatenated with the future goal points $g^{t+j}$ for every future timestep $t + j$ and is used as inputs to the decoder. The decoder then outputs the future ego-motion vectors $em^{t+j}$ for every future timestep $t + j$.

We investigated two variants of STEMM, which differ in their final output. In the first variant, STEMM outputs the ego-vehicle's future odometry, i.e., the speed and yaw rate for every future timestep: $em^{t+j} = (s^{t+j}, y^{t+j})$. Thus, we set the dimensionality $d_{em} = 2$ for the future ego-motion vector $em^{t+j} \in \mathbb{R}^{d_{em}}$. Hereby, STEMM directly replaces the use of precise temporal future odometry information, which has been used as input in our preliminary work [11]. Moreover, STEMM is trained separately in a supervised manner on the true future odometry. In the second variant, STEMM outputs a feature embedding of arbitrary size, which is directly used by BA-PTP. Hereby, BA-PTP and STEMM have to be trained jointly end-to-end and no auxiliary supervised loss on the ego-motion output $em^{t+j}$ is applied. We evaluated both variants in Section 4.

### 3.2. Loss Function

Our method is trained in a supervised manner. As the loss function for training BA-PTP, we use the root mean squared error (RMSE) between the predicted bounding boxes $\hat{b}$ and the ground truth $b$ for each of the $I$ training samples with prediction horizon $m$:

$$\text{RMSE} = \sqrt{\frac{1}{I} \sum_{i=1}^{I} \sum_{j=1}^{m} \|b_i^{t+j} - \hat{b}_i^{t+j}\|^2}. \tag{2}$$

For training of the first variant of STEMM, we also used RMSE. However, we applied separate loss functions to the outputs of STEMM, i.e., speed and yaw rate, with $\lambda$ being the loss weight for the yaw rate term:

$$\mathcal{L}_{\text{STEMM}} = \text{RMSE}_{\text{speed}} + \lambda \, \text{RMSE}_{\text{yawrate}}. \tag{3}$$

### 4. Experiments

In this Section, we evaluated our method BA-PTP on two datasets for pedestrian behavior prediction. We investigated which behavioral features contribute most to the trajectory prediction performance and verified the effectiveness of our independent encoding strategy based on ablation studies. Moreover, we demonstrated the benefit of integrating STEMM into a trajectory prediction method by comparing BA-PTP with state-of-the-art methods.

### 4.1. Datasets

We performed experiments on two different datasets:

1. PIE [2]

   PIE contains on-board camera videos recorded at 30 fps with a resolution of $1920 \times 1080$ px during daytime in Toronto, Canada. Annotations include pedestrian bounding boxes with track IDs and occlusion flags, as well as vehicle information such as speed and yaw from an on-board diagnostics sensor. The dataset contains

1842 pedestrians, divided into train, validation, and test sets by ratios of 50%, 10% and 40%, respectively.

2. ECP-Intention

The ECP-Intention dataset was created by selecting recordings collected for the Euro-City Persons (ECP) dataset [43]. The data were recorded with a two-megapixel camera (1920 × 1024 px) mounted behind the windshield at 20 fps in 31 different cities in Europe. Two-hundred-and-twenty-seven sequences from 27 cities with an average length of 16 s were selected. Each sequence was manually labeled at 5 fps (every 4th image), providing pedestrian bounding boxes with track IDs and occlusion/truncation flags, as well as head bounding boxes. Additionally, for all pedestrians, their body and head orientation relative to the line of sight of the camera in the range $[0°, 360°)$ was annotated. A value of $0°$ corresponds to an orientation directly towards the camera. For each image, odometry information is provided, captured by the vehicle's sensors and an additional IMU/GNSS sensor: the vehicle's speed and yaw rate.

Overall, ECP-Intention consists of 17,299 manually labeled images with 133,030 pedestrian bounding boxes containing 6344 unique pedestrian trajectories. The data were split into train, validation, and test subsets by ratios of 55%, 10%, and 35%, respectively. The dataset has not been published yet.

Since there were images at 20 fps available for ECP-Intention, we extended the 5 fps manual annotations to 20 fps by linear interpolation between two hand-labeled frames, called keyframes. This increased the amount of data in the observation horizon and enabled us to better train our method since it can rely on more information for encoding the motion history. In addition to that, we can generate more data samples.

### 4.2. Sampling of Future Goal Points

Section 3.1.2 explained how future goal points are sampled from the intended route of the ego-vehicle based on a constant velocity model. ECP-Intention as well as PIE do not provide data on the intended route from the navigation system itself. Therefore, we calculated the driven route based on the odometry information available in both datasets as an approximation of the intended route. This driven route only provides spatial information for goal point sampling. Precise temporal information was discarded.

### 4.3. Implementation

We used Gated Recurrent Units (GRUs) with 256 hidden units and *tanh* activation for the encoding streams as well as for the decoder. Observation ($n$) and prediction horizon ($m$) were set based on the dataset used: For PIE, we followed [2] and used 0.5 s of observations ($n = 15$) and predicted 1.5 s ($m = 45$). We sampled tracks with a sliding window approach and a stride of 7. For ECP-Intention, we observed 0.6 s ($n = 13$) and predicted 1.6 s ($m = 32$). Tracks were sampled with a stride of 2. If there were missing data frames within a sequence of the same pedestrian (e.g., due to occlusion), we split the sequence into multiple tracks to ensure consecutive tracks during training and testing. We trained every model for 100 epochs using the Adam [44] optimizer with a batch size of 128 and $L_2$ regularization of $5 \times 10^{-3}$ for ECP-Intention and $L_2$ regularization of $10^{-3}$ for PIE. After each temporal attention block for both BA-PTP and STEMM, we used a dropout of 0.5, following [15]. If not stated otherwise, the initial learning rate for ECP-Intention and PIE was set to $5 \times 10^{-4}$ and $10^{-3}$, respectively. If the validation loss had not improved for five epochs during training, the learning rate was reduced by a factor of 5.

Since we did not have annotated poses for ECP-Intention and no annotations for behavioral features at all for PIE, we needed to generate the data beforehand. To infer the pose and body orientation of pedestrians, we used the following architecture: For the detection and pose estimation of pedestrians, we used VRU-Pose SSD [45], extended by the orientation loss used in Pose-RCNN [46]. In this way, we simultaneously predicted the pose and body orientation of pedestrians. The model was trained on a combination of three datasets: ECPDP [47], TDUP [48], and additional non-public data. To associate

these detections with the ground truth boxes of ECP-Intention and PIE, we matched every ground truth bounding box with the detection with the highest *Intersection over Union* (*IoU*). Missing attribute values are ignored by our method using a masking layer.

### 4.4. Data Preparation

For the trajectory inputs $BB_{obs}$ and $HB_{obs}$, we used the ground truth bounding boxes. The reason for that is to make the trajectory prediction independent of a pedestrian detector's performance as in the majority of previous works [2,4,5,10,14]. We normalized $BB_{obs}$ and $HB_{obs}$ by subtracting the first bounding box from the whole track, i.e., converting the absolute to relative pixel coordinates, as is the case in [2].

The pedestrian's pose $p_i$ was normalized by dividing each x-coordinate by the image width and each y-coordinate by the image height. The values for body and head orientation wee normalized to the range $[0, 1)$. For odometry information, we did not apply any pre-processing.

For normalization of the goal points for STEMM, we used the ego-centric coordinate system of the ego-vehicle at the final observed timestep $t$ as the reference system. We obtained goal points within that reference system using common coordinate transformations. We used the front-left-up convention, meaning the x-axis points towards the front for the ego-vehicle. Therefore, the y-coordinate of the goal points was zero, as long as the intended route was straight. For normalization of the past ego-vehicle positions, we simply used the ego-centric coordinate system at the first observed timestep $t - n$.

### 4.5. Metrics

We used the standard metrics [2–4] for evaluation on both datasets, PIE and ECP-Intention: the mean squared error (MSE) of the bounding boxes' upper-left and lower-right corner was calculated over the different prediction horizons as well as the MSE of the bounding box centers over the whole prediction horizon ($C_{MSE}$) and only for the final timestep ($CF_{MSE}$). For ECP-Intention, we calculated the metrics only for the keyframes to evaluate the performance only on the hand-labeled data. To this end, we restricted test samples to end with keyframes during data generation enabling us to measure prediction performance exactly 1.6 s into the future.

We averaged the trajectory prediction results of all models over four different experiment runs. In the following sections, we report the average performance with standard deviation in integer pixel errors by mathematical rounding.

### 4.6. Results on the ECP-Intention Dataset

In this subsection, we performed experiments on the ECP-Intention dataset. Initially, we investigated the relevance of different behavioral features. In the following, we demonstrate the effectiveness of independently encoding multiple input features. Finally, we evaluated the benefit of integrating STEMM into BA-PTP and compared it with the state-of-the-art methods.

#### 4.6.1. Relevance of Different Behavioral Features

We performed an ablation study on ECP-Intention to investigate which behavioral features contribute most to the prediction performance. In order to isolate the effect of specific behavioral features, we decoupled the camera's ego-motion from the trajectory prediction task by utilizing ground truth future odometry, i.e., speed and yaw rate, for these experiments instead of inferring the future ego-motion using STEMM. We compared our results to $PIE_{traj+speed}$, which is an extension of $PIE_{traj}$ [2] that additionally uses ground truth ego-vehicle speed information. The results of our method when we use only a subset of the behavioral features are shown in Table 2.

**Table 2.** Ablation study on the ECP-Intention dataset (BB = Body Bounding Box, HB = Head Bounding Box, BO = Body Orientation, HO = Head Orientation, P = Pose). * denotes no use of yaw rate.

| | Model | MSE Avg (Std) | | $C_{MSE}$ Avg (Std) | $CF_{MSE}$ Avg (Std) |
|---|---|---|---|---|---|
| | | 0.8 s | 1.6 s | 1.6 s | 1.6 s |
| 1 | $PIE_{traj+speed}$ [2] * | $417 \pm 10$ | $2120 \pm 63$ | $1997 \pm 67$ | $6516 \pm 235$ |
| 2 | BA-PTP$_{BB-Y}$ * | $392 \pm 12$ | $2078 \pm 26$ | $1982 \pm 82$ | $6510 \pm 110$ |
| 3 | BA-PTP$_{BB}$ | $242 \pm 1$ | $917 \pm 10$ | $830 \pm 10$ | $2510 \pm 38$ |
| 4 | BA-PTP$_{BB+HB}$ | $211 \pm 4$ | $801 \pm 10$ | $716 \pm 11$ | $2204 \pm 24$ |
| 5 | BA-PTP$_{BB+HB+HO}$ | $205 \pm 2$ | $731 \pm 9$ | $652 \pm 9$ | $1953 \pm 24$ |
| 6 | BA-PTP$_{BB+HB+BO}$ | $206 \pm 2$ | $734 \pm 16$ | $656 \pm 13$ | $1957 \pm 74$ |
| 7 | BA-PTP$_{BB+HB+P}$ | $197 \pm 1$ | $679 \pm 5$ | $597 \pm 5$ | $1788 \pm 17$ |
| 8 | BA-PTP$_{BB+HB+BO+HO}$ | $201 \pm 3$ | $716 \pm 14$ | $637 \pm 12$ | $1920 \pm 48$ |
| 9 | BA-PTP$_{BB+HB+HO+P}$ | $193 \pm 5$ | $641 \pm 13$ | $568 \pm 11$ | $1658 \pm 30$ |
| 10 | BA-PTP$_{BB+HB+BO+P}$ | $193 \pm 5$ | $635 \pm 14$ | $565 \pm 13$ | $1631 \pm 37$ |
| 11 | **BA-PTP$_{BB+HB+BO+HO+P}$** | $\mathbf{187 \pm 2}$ | $\mathbf{615 \pm 17}$ | $\mathbf{544 \pm 15}$ | $\mathbf{1579 \pm 65}$ |
| 12 | BA-PTP$_{BB+HB+infer.\ BO}$ | $205 \pm 4$ | $741 \pm 18$ | $659 \pm 16$ | $1986 \pm 63$ |
| 13 | BA-PTP$_{BB+HB+infer.\ BO+P}$ | $194 \pm 5$ | $656 \pm 25$ | $579 \pm 22$ | $1703 \pm 86$ |

One initial finding is that our method outperforms $PIE_{traj+speed}$ on all metrics by a significant margin even without incorporating any behavioral features into the prediction process (BA-PTP$_{BB}$). We attribute this to the utilization of the yaw rate of the ego-vehicle, which $PIE_{traj+speed}$ does not use, particularly enhancing prediction performance in turning scenarios. A performance comparison with our model that excludes the use of the yaw rate (BA-PTP$_{BB-Y}$) validates this assumption. Upon incorporating the yaw rate to our model, we observed a notable performance increase of 38% for MSE-0.8 s and up to 55% for longer-term predictions (MSE-1.6 s). These findings suggest that integrating the ego-vehicle's yaw rate is crucial for effectively accounting for the ego-motion of the camera in the sequences of ECP-Intention.

Furthermore, we observed that adding the head bounding boxes of pedestrians, in addition to their body bounding boxes, is beneficial. Adding this information to BA-PTP results in an increase of performance by 12–13% across all metrics (BA-PTP$_{BB+HB}$). This confirms our motivation of adding head bounding boxes to obtain a more stable encoding of pedestrian trajectories and to reduce the box wobbling effects. In the rest of this work, we refer to the combination of body and head bounding boxes as the past trajectory.

Rows 5–7 illustrate the performance variation when utilizing only one of the behavioral features (BO, HO, P) alongside the past trajectory. Every individual feature contributes to the enhancement of trajectory prediction. Among the behavioral features, the pose shows the greatest impact on the prediction performance, resulting in an improvement of 6% for MSE-0.8 s and 15% for MSE-1.6 s when compared to solely using the past trajectory. Comparing the improvements in short-term (0.8 s) and long-term (1.6 s) predictions indicates that the advantage of incorporating behavioral features grows as the prediction horizon becomes longer.

The results of our method when integrating multiple behavioral features in combination are shown in rows 8–11 and allow us to derive several conclusions. The combination of two behavioral features yields an improved performance compared to using a single additional feature, suggesting a complementary relationship between these features. Utilizing all behavioral features, i.e., body orientation, head orientation, and pose (BA-PTP$_{BB+HB+BO+HO+P}$) attains the most accurate prediction of future trajectories of pedestrians on the ECP-Intention dataset. The prediction performance increases by 23% in MSE-1.6 s, 24% in $C_{MSE}$, and 28% in $CF_{MSE}$ compared to only using the past trajectory. This demonstrates the advantage of incorporating behavioral features of pedestrians in pedestrian trajectory prediction.

Experiments with using inferred body orientation instead of the ground truth annotations as input for both training and inference are shown in the last two rows of Table 2. We noticed a slight performance decrease across all metrics when compared to the models using ground truth annotations as input, which can be explained by the introduced noise from using inferred values. Nevertheless, notable improvements are observed in comparison to solely relying on bounding boxes.

### 4.6.2. Effect of Independent Encoding Streams

We conducted a comparison between our attention-based modality fusion (*Independent*) and a variant of the model in which all input features were concatenated first and processed using a single encoding stream (*Concat*) in Table 3. Hence, this model variant solely relies on temporal attention and does not incorporate modality attention. Also, we did not change the number of hidden units for this ablation. The only additional change we made is that we initialized the decoder's initial state with zeros, which we found to result in the best performance for the *Concat* encoding strategy. We show this ablation study for two models: BA-PTP$_{BB+HB+BO+P}$ and BA-PTP$_{BB+HB+BO+HO+P}$.

**Table 3.** Comparison of different encoding strategies on the ECP-Intention dataset (C = Concat, I = Independent).

| Model | MSE Avg (Std) | | $C_{MSE}$ Avg (Std) | $CF_{MSE}$ Avg (Std) |
|---|---|---|---|---|
| | 0.8 s | 1.6 s | 1.6 s | 1.6 s |
| BA-PTP$_{BB+HB+BO+P}$-C | $367 \pm 7$ | $989 \pm 23$ | $895 \pm 19$ | $2424 \pm 84$ |
| **BA-PTP$_{BB+HB+BO+P}$-I** | **$193 \pm 5$** | **$635 \pm 14$** | **$565 \pm 13$** | **$1631 \pm 37$** |
| BA-PTP$_{BB+HB+BO+HO+P}$-C | $371 \pm 9$ | $996 \pm 20$ | $907 \pm 16$ | $2406 \pm 65$ |
| **BA-PTP$_{BB+HB+BO+HO+P}$-I** | **$187 \pm 2$** | **$615 \pm 17$** | **$544 \pm 15$** | **$1579 \pm 65$** |

Our *Independent* encoding strategy outperforms the *Concat* encoding strategy in both models, yielding superior results. For example, for BA-PTP$_{BB+HB+BO+HO+P}$ MSE-1.6 s, $C_{MSE}$ and $CF_{MSE}$ improve by 38%, 40%, and 34%, respectively. Moreover, for the two models using the *Concat* encoding strategy, the additional use of the head orientation does not improve the prediction performance in contrast to the *Independent* encoding strategy. This also indicates the benefit of using independent encoding streams for different behavioral features. This result may be caused by the higher model complexity in terms of the number of weights for the *Independent* strategy with a fixed number of hidden units in the GRUs.

### 4.6.3. Results Using STEMM

We then performed experiments on the ECP-Intention dataset, evaluating our method BA-PTP using ego-motion predictions from STEMM for testing. The pre-trained variant of STEMM has been used for ECP-Intention, as it leads to better performance compared to the end-to-end variant. This means that STEMM directly predicts the speed and yaw rate of the ego-vehicle for every future timestep and replaces the use of precise temporal future odometry information from Section 4.6.1. The better performance of the pre-trained variant of STEMM for ECP-Intention can be explained by our findings from Section 4.6.1. We observed significant performance gains when adding the yaw rate of the ego-vehicle to the inputs, indicating its importance. Pre-training STEMM on the true future odometry seems to satisfy the yaw rate requirement more effectively compared to a latent feature vector in the end-to-end variant without an explicit supervised yaw rate loss. For training STEMM, we set the initial learning rate to $5 \times 10^{-3}$ and used $L_2$ regularization of $10^{-4}$. Furthermore, the loss weight $\lambda$ for the yaw rate was set to 50. The results are shown in Table 4.

**Table 4.** Comparison with SOTA on the ECP-Intention dataset (BB = Body Bounding Box, HB = Head Bounding Box, BO = Body Orientation, HO = Head Orientation, P = Pose). * denotes using ground truth future odometry. † denotes using no odometry information.

| Model | MSE Avg (Std) | | $C_{MSE}$ Avg (Std) | $CF_{MSE}$ Avg (Std) |
|---|---|---|---|---|
| | 0.8 s | 1.6 s | 1.6 s | 1.6 s |
| $PIE_{traj}$ [2] † | $434 \pm 24$ | $2216 \pm 71$ | $2088 \pm 70$ | $6785 \pm 136$ |
| $PIE_{traj+speed}$ [2] * | $417 \pm 10$ | $2120 \pm 63$ | $1997 \pm 67$ | $6516 \pm 235$ |
| SGNet [4] † | $390 \pm 8$ | $2137 \pm 94$ | $2033 \pm 92$ | $6708 \pm 330$ |
| BA-PTP$_{BB+HB}$ † | $369 \pm 14$ | $2018 \pm 24$ | $1912 \pm 23$ | $6382 \pm 163$ |
| BA-PTP$_{BB+HB}$ | $229 \pm 4$ | $991 \pm 11$ | $903 \pm 11$ | $2882 \pm 27$ |
| **BA-PTP$_{BB+HB+BO+HO+P}$** | $\mathbf{207 \pm 2}$ | $\mathbf{824 \pm 16}$ | $\mathbf{751 \pm 14}$ | $\mathbf{2339 \pm 53}$ |
| BA-PTP$_{BB+HB}$ * | $211 \pm 4$ | $801 \pm 10$ | $716 \pm 11$ | $2204 \pm 24$ |
| BA-PTP$_{BB+HB+BO+HO+P}$ * | $187 \pm 2$ | $615 \pm 17$ | $544 \pm 15$ | $1579 \pm 65$ |

When replacing precise ground truth information with ego-motion predictions from STEMM in BA-PTP, we observed a decreasing performance across all metrics. For BA-PTP$_{BB+HB}$, the prediction error increases by 8% for MSE-0.8 s and for a longer prediction horizon by up to 23% for MSE-1.6 s. However, comparing to our ablation, which does not use any odometry information (BA-PTP$_{BB+HB}$†), underlines the benefit of integrating STEMM into BA-PTP. The integration of STEMM results in significant improvements, i.e., a lower MSE-1.6 s of 50% and a lower $C_{MSE}$ of 52%.

We then compared our behavior-aware model (BA-PTP$_{BB+HB+BO+HO+P}$) with the current state-of-the-art for pedestrian trajectory prediction in the image plane: PIE$_{traj}$ [2] and SGNet [4]. The latter is currently the best-performing model on the PIE dataset. As we already discussed in Section 4.6.1, the baselines achieve bad prediction results on the ECP-Intention dataset as they do not utilize information about the yaw rate. We have surpassed SGNet's performance by a large margin. In terms of MSE-1.6 s, $C_{MSE}$, and $CF_{MSE}$, we outperformed them by 61%, 63% and 65%, respectively.

### 4.7. Results on the PIE Dataset

In this section, we evaluated our proposed method on the PIE dataset [2] using inferred values for body orientation and pose of pedestrians, as described in Section 4.3. For our experiments on the PIE dataset, we additionally compared our results to PIE$_{full}$ [2], Action-Aware Enc-Dec Network [9], and Crossmodal Transformer [8]. PIE$_{full}$ incorporates the estimated pedestrian's intention to cross as well as predicted speed of the ego-vehicle for the bounding box predictions. As described in Section 2, the latter two methods also use behavioral features of pedestrians and [9] even utilizes ground truth future ego-vehicle odometry. The results for the baselines were taken from the respective publications.

Similar to our ablations in Section 4.6.1, we first used ground truth future odometry, i.e., speed, for these experiments, to decouple the influence of ego-motion prediction accuracy on the prediction performance. As there are no annotations for the head bounding boxes for the PIE dataset, we could not investigate this in our experiments.

Table 5 shows that using inferred behavioral features in addition to bounding boxes improves the prediction of pedestrians' future trajectories also on the PIE dataset. Compared to only using bounding boxes (BA-PTP$_{BB}$*), adding body orientation increases the prediction performance by a small margin (1% in MSE-1.5 s), while using pose information improves the performance significantly (11% in MSE-1.5 s). The best average prediction performance is achieved by BA-PTP$_{BB+BO+P}$*. We observed a few points of improvement across all metrics when comparing to BA-PTP$_{BB+P}$*. In terms of MSE-1.5 s, $C_{MSE}$, and $CF_{MSE}$, we outperformed Action-Aware Enc-Dec Network—which also uses ground truth odometry information—by 13%, 16%, and 15%, respectively.

**Table 5.** Trajectory prediction results on the PIE dataset (BB = Body Bounding Box, BO = Body Orientation, P = Pose). * denotes using ground truth future odometry. † denotes using no odometry information.

| Model | MSE Avg (Std) | | | $C_{MSE}$ Avg (Std) | $CF_{MSE}$ Avg (Std) |
|---|---|---|---|---|---|
| | 0.5 s | 1 s | 1.5 s | 1.5 s | 1.5 s |
| PIE$_{traj}$ [2] † | 58 | 200 | 636 | 596 | 2477 |
| PIE$_{full}$ [2] | - | - | 559 | 520 | 2162 |
| PIE$_{traj+speed}$ [2] * | 60 ± 2 | 173 ± 4 | 498 ± 9 | 450 ± 8 | 1782 ± 40 |
| Action-Aware Enc-Dec Network [9] * | 43 | 160 | 457 | 436 | 1683 |
| Crossmodal Transformer [8] | 43 | 149 | 443 | 413 | 1670 |
| SGNet [4] † | **34** | **133** | 442 | 413 | 1761 |
| BA-PTP$_{BB}$ † | 53 ± 1 | 188 ± 3 | 615 ± 11 | 580 ± 11 | 2469 ± 44 |
| BA-PTP$_{BB}$ * | 48 ± 1 | 154 ± 2 | 459 ± 4 | 429 ± 4 | 1746 ± 24 |
| BA-PTP$_{BB+BO}$ * | 50 ± 1 | 155 ± 1 | 452 ± 5 | 421 ± 5 | 1692 ± 34 |
| BA-PTP$_{BB+P}$ * | 51 ± 1 | 146 ± 2 | 405 ± 6 | 376 ± 5 | 1473 ± 31 |
| BA-PTP$_{BB+BO+P}$ * | 50 ± 1 | 143 ± 3 | 395 ± 4 | 366 ± 4 | 1421 ± 9 |
| BA-PTP$_{BB}$ | 47 ± 0 | 158 ± 4 | 495 ± 22 | 463 ± 21 | 1940 ± 105 |
| BA-PTP$_{BB+BO+P}$ | 46 ± 0 | 137 ± 1 | **411 ± 4** | **381 ± 3** | **1593 ± 22** |

Since we investigated the influence of including behavioral features for pedestrian trajectory predicted in an idealized setting using ground truth future odometry information, we then evaluated the performance of BA-PTP when it was trained in an end-to-end manner alongside STEMM for the purpose of ego-motion compensation. The end-to-end variant of STEMM has been used for PIE as it leads to better performance compared to the pre-trained variant. We explain this by the fact that the scenes in the PIE dataset mostly cover scenarios where the ego-vehicle drives in a straight line or stands still, in contrast to ECP-Intention. In Section 4.6.3, the pre-trained variant of STEMM performs better on the ECP-Intention dataset due to explicitly predicting the yaw rate of the ego-vehicle. This seems to be less important on the PIE dataset, where turning scenarios are underrepresented. For PIE, the latent features learned by the end-to-end variant of STEMM provide better means for ego-motion compensation to be used by BA-PTP. We changed the initial learning rate to $5 \times 10^{-4}$ when using STEMM. The dimensionality of the future ego-motion vectors $em^{t+j}$ is $d_{em} = 64$. The results are shown in the final two rows of Table 5.

When we incorporated STEMM into BA-PTP instead of using precise future odometry information, the prediction performance drops slightly. For BA-PTP$_{BB+BO+P}$ we lose 4% in MSE-1.5 s and $C_{MSE}$ and 12% in $CF_{MSE}$. However, using STEMM's output instead of ground truth odometry results in a better performance for shorter prediction horizons, i.e., four points for 0.5 s and six points for 1 s. Moreover, when comparing BA-PTP$_{BB}$ with its counterpart, which neglects any odometry information (BA-PTP$_{BB}$†), we observed a highly increasing performance across all metrics. Additionally, BA-PTP$_{BB}$ outperforms PIE$_{full}$—which also utilizes ego-motion predictions—by a large margin (11% in MSE-1.5 s). This aligns with our hypothesis that the latent features provided by the end-to-end variant of STEMM allow for more effective ego-motion compensations on PIE compared to explicit ego-motion predictions like speed, which is predicted by PIE$_{full}$. These findings demonstrate the advantage of integrating STEMM into our pedestrian trajectory prediction method.

Comparing to the previous best-performing approaches for pedestrian trajectory prediction shows that we outperform Crossmodal Transformer—which also uses the pose and body orientation of pedestrians as input—by 7% in MSE-1.5 s. Further, we observe that SGNet, which solely relies on pedestrian bounding boxes as input, is able to better capture the short-term motions of pedestrians (0.5 s and 1 s). However, BA-PTP$_{BB+BO+P}$ achieves state-of-the-art performance in terms of MSE-1.5 s and $C_{MSE}$, as well as $CF_{MSE}$, improving over SGNet by 7%, 7% and 9%, respectively. Therefore, we conclude that the importance

of ego-motion compensation and the benefit of behavioral features increases with higher prediction horizons. For behavioral features, this has also been shown in Section 4.6.1.

*4.8. Qualitative Results*

Figure 4 shows the qualitative results of our proposed models on ECP-Intention and on PIE. Due to the difference in the used inputs for our models for ECP-Intention and PIE, we introduced two naming conventions here. First, we refer to our behavior-aware models as $BA\text{-}PTP_{beh}$, i.e., $BA\text{-}PTP_{BB+HB+BO+HO+P}$ for ECP-Intention and $BA\text{-}PTP_{BB+BO+P}$ for PIE. Second, we refer to ablations of our model, which solely relies on the past trajectory, as $BA\text{-}PTP_{traj}$, i.e., $BA\text{-}PTP_{BB+HB}$ for ECP-Intention and $BA\text{-}PTP_{BB}$ for PIE. We compared our behavior-aware model $BA\text{-}PTP_{beh}$ with three other models:

- An ablation of our model solely relying on the past trajectory and not using any odometry information, i.e., $BA\text{-}PTP_{traj}$†;
- $BA\text{-}PTP_{traj}$, i.e., our method without using behavioral features;
- $PIE_{traj+speed}$.

Images (a) and (b) depict two example situations in which the ego-vehicle is executing a turning maneuver. In these scenarios, both of our ego-motion-aware models effectively account for the camera's ego-motion due to the incorporation of our newly proposed STEMM, thereby compensating for its impact, whereas the two other models lag behind. $BA\text{-}PTP_{traj}$† does not incorporate any odometry information and $PIE_{traj+speed}$ is missing the ability to compensate for the horizontal movement of the image due to neglecting yaw rate information. These two examples underline the effectiveness of integrating STEMM into a pedestrian trajectory prediction model. However, in image (g) we show a failure case on the PIE dataset, where the ego-vehicle is also making a turn. We still see that the ego-motion-aware models perform better in this scenario and predict the turn of the ego-vehicle, but they cannot completely compensate for the ego-motion of the camera.

Examples (c) and (e) illustrate instances where $BA\text{-}PTP_{beh}$ successfully predicts a change in motion, specifically that the pedestrians depicted in both images will start crossing the street. On the contrary, all other models fail to accurately estimate the correct future behavior in these cases. This observation can be attributed to the pedestrians' orientations towards the street, highlighting the potential improvement in pedestrian trajectory prediction by leveraging behavioral features such as body and head orientation as well as pose. Moreover, we show a scenario on the ECP-Intention dataset, where a pedestrian suddenly stops crossing the street, in image (d). All models correctly predict that the pedestrian will not cross the street. Additionally, $BA\text{-}PTP_{beh}$ performs best in compensating for the ego-motion of the vehicle surpassing $PIE_{traj+speed}$, which uses ground truth future speed information.

In the image (f), we also show a failure case, where a pedestrian decides to cross the street in high traffic. None of the models predict this behavior. $BA\text{-}PTP_{beh}$ even estimates the future position on the sidewalk. This failure might be explained by the missing multimodality in BA-PTP's output as this is a demonstrative scenario where two possible trajectories are plausible and the model cannot predict both. Image (h) displays a pedestrian walking across a zebra crossing, exhibiting a linear motion pattern. All models accurately predict the future trajectory in this case, as the positional information from past bounding boxes, which is utilized by all models, proves to be sufficient.

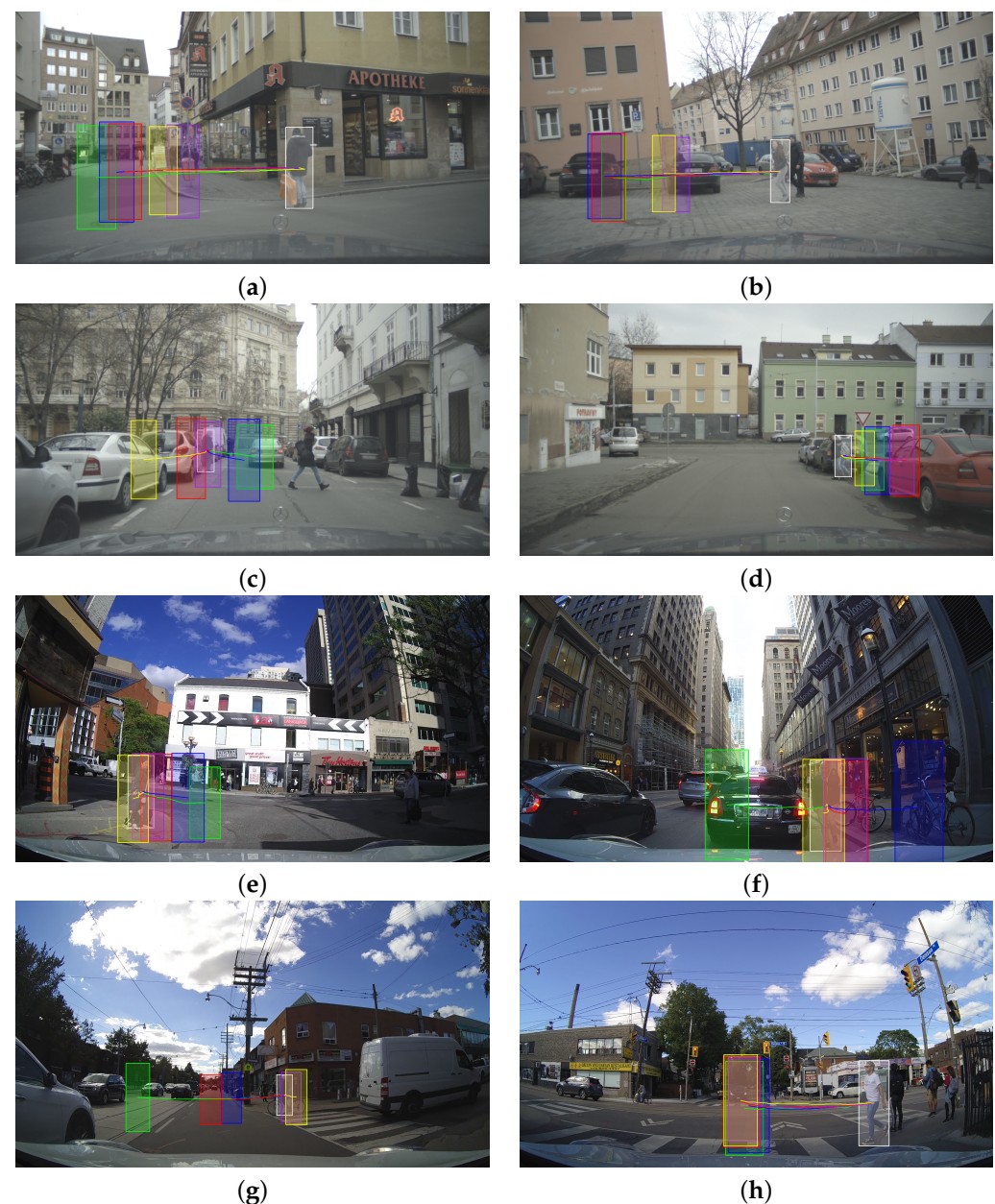

**Figure 4.** Qualitative results on the ECP-Intention dataset (images **a**–**d**) and on the PIE dataset (images **e**–**h**). The images show the last observed timestep with the last observed bounding box (white box) as well as predictions into the future (1.6 s for ECP-Intention and 1.5 s for PIE). Visualized are the final predicted bounding box and a line connecting the centers of the predicted bounding boxes for the intermediate timesteps. Ground truth is displayed in green. Predictions of $PIE_{traj+speed}$ in yellow, our model solely relying on the past trajectory and not using any odometry information $BA\text{-}PTP_{traj}$† in purple, $BA\text{-}PTP_{traj}$ in red and our behavior-aware model $BA\text{-}PTP_{beh}$ in blue.

## 5. Discussion

This study demonstrated the benefit of using behavioral features in a pedestrian trajectory prediction method. It also showcased that extending the representation of pedestrian trajectories by head bounding boxes in addition to solely using body bounding boxes improves the prediction performance by learning better encodings of the motion history of pedestrians. Moreover, we have shown the effectiveness of including STEMM in our method to account for the ego-motion of the camera in ego-centric views. BA-PTP outperformed all prior state-of-the-art works on two datasets for pedestrian behavior prediction.

Compared to [11], we extended our preliminary work by including information about the head position of pedestrians as well as adapting the bounding box representation to reduce the box wobbling effects. One major limitation of our previous work was the use of ground truth future odometry information. In this work, we introduced STEMM and integrated it into BA-PTP, dropping the need for highly precise ego-motion information from the future. This makes our extended method usable in a real-world application with unknown future odometry. During the methodical extension of BA-PTP, we also had to revisit the choice of hyperparameters used to train the model (cf. Section 4.3) to cope with the new changes. In order to ensure fair quantitative comparisons, we compared our results using ground truth future odometry (Tables 2 and 5) with our preliminary results reported in [11]. On both datasets, ECP-Intention and PIE, we observed remarkable improvements in the prediction performance for our extended version of our method. For the model variant of BA-PTP that solely utilizes the past body bounding boxes of pedestrians, we achieved an improvement of 5% in terms of MSE-1.5 s on PIE and 6% in terms of MSE-1.6 s on ECP-Intention. This can be partly attributed to the optimized hyperparameters, but particularly to the change in representing bounding boxes (Section 3), which contributes to reducing the box wobbling effects. When comparing the results of the best performing behavior-aware models, we note even more significant gains in terms of performance. On the ECP-Intention dataset, we outperformed our method from [11] by 19% across all metrics. The even greater improvement in performance indicates the advantage of incorporating the head bounding boxes, which were not utilized in our previous method. This extension results in a more stable encoding of pedestrian trajectories in the image plane. Also, this aligns with our hypothesis from Section 3. Another notable insight is the superior performance of our extended method BA-PTP using the ego-motion predictions from STEMM (cf. BA-PTP$_{BB+BO+P}$ in Table 5) against our results reported in [11] in terms of MSE-0.5 s, MSE-1 s, and MSE-1.5 s on the PIE dataset. Despite the utilization of ground truth future odometry information in [11], we still improved over our preliminary work, highlighting the contribution of STEMM to improving pedestrian trajectory prediction in the image plane.

We compared our proposed method to current state-of-the-art pedestrian trajectory prediction methods in the image plane in Section 4. The enhancement observed in comparison to [8] further demonstrates the capabilities of BA-PTP. Both methods share similar inputs, namely bounding boxes, poses, and the body orientation of pedestrians as well as the ego-motion of the vehicle. However, the crossmodal transformer based encoders in [8] are used to learn how the pedestrian modalities (box and pose) are influenced by the *past ego-vehicle odometry information*, whereas BA-PTP explicitly uses the *ego-motion predictions* provided by STEMM. This suggests that the integration of STEMM for ego-motion compensation helps to predict more accurate future bounding boxes. Additionally, only the final observed body orientation of a pedestrian was incorporated in [8] to encode the basic state of the pedestrian, in contrast to our method, which encodes a sequence of observed past body orientations.

We have also demonstrated superior performance compared to the method presented in [9], which also uses the pose of pedestrians in addition to bounding boxes as input. We attribute its lower performance to its encoding strategy, where the input modalities are concatenated first before being processed jointly by LSTM blocks contrary to our independent encoding approach. The benefit of using independent encoding streams for different input modalities has been shown for our method in Section 4.6.2. Moreover, Ref. [9] further utilizes ground truth future odometry information to compensate for the ego-motion of the camera, whereas BA-PTP utilizes ego-motion predictions provided by STEMM.

Our study has several limitations. In order to deploy STEMM in a real-world application, it is necessary to know the intended route of the ego-vehicle, e.g., provided by a navigation system. Nonetheless, in the domain of automated vehicles, knowing the intended route of the ego-vehicle can be considered given. The benefit and importance

of accounting for the ego-motion of the camera for pedestrian trajectory prediction in the image plane has been demonstrated in Sections 4.6.3 and 4.7.

Despite the state-of-the-art performance of BA-PTP, the quantitative as well as qualitative evaluations indicate that there is still room for improvement. Due to the highly variable behavior of pedestrians, we can observe scenarios in which multiple future trajectories of pedestrians are plausible. BA-PTP's prediction performance is limited in such cases, because it is missing the capability of predicting multimodal trajectories. However, this fundamental extension is beyond the scope of this study and is left for future work. Furthermore, we focused on extracting the behavioral features from camera images and explicitly utilizing them with independent encoding streams in this work. An alternative to this approach would have been to directly use the observed images or image crops as input to a trajectory prediction method in an end-to-end manner. Future research directions should consider investigations regarding the effectiveness of such an approach.

## 6. Conclusions

In this work, we presented BA-PTP, a novel approach to pedestrian trajectory prediction from the ego-vehicle camera perspective. Behavioral features extracted from visual observations, such as the body and head orientation of pedestrians as well as their pose, were utilized by BA-PTP in addition to positional information from body and head bounding boxes. We added the head bounding boxes to the positional information to provide a more stable encoding of pedestrian trajectories in the image plane. To enhance the learned embeddings of pedestrians' motion history, we employed independent encoding streams for each input modality and combined the resulting outputs. By adopting this approach, we explicitly leveraged the information provided by pedestrians regarding their intended movement. To overcome the challenge of ego-motion compensation for pedestrian trajectory prediction from the ego-vehicle camera perspective, we integrated STEMM, a novel approach for ego-motion prediction. It uses spatial goal points that are sampled from the intended route of the ego-vehicle and replaces the use of precise temporal future ego-vehicle odometry information.

By evaluating BA-PTP on two datasets for predicting pedestrian behavior, we showed that including behavioral features benefits pedestrian trajectory prediction and demonstrated the strength of incorporating STEMM into our method. Further, we have achieved a state-of-the-art performance on the PIE dataset, outperforming prior work by 7% in MSE-1.5 s and $C_{MSE}$, as well as by 9% in $CF_{MSE}$.

**Author Contributions:** Conceptualization, P.C. and M.B.; methodology, P.C.; software, P.C.; investigation, P.C.; resources, U.K.; data curation, P.C.; writing—original draft preparation, P.C.; writing—review and editing, M.B., U.K. and B.Y.; visualization, P.C.; supervision, M.B. and B.Y.; funding acquisition, U.K. All authors have read and agreed to the published version of the manuscript.

**Funding:** This work is a result of the research project STADT:up—Solutions and Technologies for Automated Driving in Town: An urban mobility project. The project is supported by the Federal Ministry for Economic Affairs and Climate Action (BMWK), based on a decision taken by the German Bundestag. The authors are solely responsible for the content of this publication.

**Data Availability Statement:** Two datasets were used in this work. PIE is publicly accessible at https://data.nvision2.eecs.yorku.ca/PIE_dataset/ (accessed on 31 July 2023). The ECP-Intention dataset is not published yet.

**Acknowledgments:** We thank Sebastian Krebs for his valuable support throughout the research in this work. We thank Arij Bouazizi and Julian Wiederer for proofreading the manuscript and providing constructive feedback.

**Conflicts of Interest:** The authors declare no conflict of interest. The funders had no role in the design of the study; in the collection, analyses, or interpretation of data; in the writing of the manuscript; or in the decision to publish the results.

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
