# Peer review of "Behavior-Aware Pedestrian Trajectory Prediction in Ego-Centric Camera Views with Spatio-Temporal Ego-Motion Estimationâ€"

_make, doi:10.3390/make5030050_

Round 1
Reviewer 1 Report
This study examined behavior-aware pedestrian trajectory prediction methods in ego-centric camera views with spatio-temporal ego-motion estimation. While this is an interesting and innovative study, its flaw in self-plagiarism may make it unacceptable for publication yet.
Although this manscript is extended from a conference paper, self-plagiarism should be avoided. The conference paper has been published already. One can not publish two papers with similar content, which can be seen as academic misconduct. The SIMILARITY rate between the two papers is as high as 43% (as provided by the journal office)!
Some other comments:
In fact, the updated methods might be not different enough from their conference paper to deserve to be published in a new paper. Many of methods and results are the same as what was reported in their conference paper. Or else, the authors should stately clearly the difference and significance in innovation in relation to their conference paper.
This study lacks a thorough discussion on their methods/results in relation to previous work (including their conference paper). E.g., why the proposed methods are better than previous ones.
Grammatical mistakes (which can be seen in the first sentence in the introduction) should be well checked.
Editing of English language required
Reviewer 2 Report
The authors of this manuscript presents the "Behavior-Aware Pedestrian Trajectory Prediction in Ego-Centric Camera Views with Spatio-Temporal Ego-Motion Estimation". However, I have some suggestions to improve this manuscript.
1. Present the previous studies in a table and highlight the contribution of authors.
2. what is the reference of equation (1) line 241?
3. It will be more attractive if you write the algorithm in section 3.
4. Discuss the findings in a seprate paragraph before the conclusion.
5. add some more references from 2023.
Reviewer 3 Report
The paper is a study on the behavior-aware pedestrian trajectory prediction in ego-centric camera views with spatio-temporal ego-motion estimation and is considered a valuable and interesting study in related fields. The reviewer's opinions are as follows.
1. Abstract should be concisely and clearly described, including the background, purpose, method, result, and conclusion of the study.
2. In the description, ambiguous expressions should be avoided and quantitative numerical values or objective grounds should be presented.
3. It is necessary to describe existing efforts(papers) regarding the problems (not the simple description of the existing studies). The methods that solved the problems perceived in previous similar studies should be described in detail(academic excellence on this paper).
4. In the section describing the simulations and experiments, the composition of the simulations and datasets should be clearly explained. It should be described in such a way that readers who related the fields can understand processes in detail. In other words, it should be possible to solve the questions by the composition of the provided experimental environments including the models suggested and datasets provided. In the simulation part, verification of the proposed methodology should be sufficiently presented.
5. In the 'conclusion' part, it is necessary to describe the limitations of the study and additional studies required in the future. It is recommended to describe the interpretation of the research results in an easy-to-understand manner.
6. Authors should ensure the quality of the paper in its overall descriptive expression and follow the format of the journal.
Thank you very much.
